# Blood Concentrations and Dietary Intake of Cd among the General Population in South Korea

**DOI:** 10.3390/ijerph19010152

**Published:** 2021-12-23

**Authors:** Chan-Seok Moon

**Affiliations:** Department of Safety and Health, College of Health Sciences, Catholic University of Pusan, Busan 46252, Korea; csmoon@cup.ac.kr; Tel.: +82-51-510-0633

**Keywords:** Cd, blood concentration, food group, decrease, dietary intake, grains and cereals, potato and starch, fruits, vegetables

## Abstract

This study aimed to identify the time trends of blood Cd concentrations and their correlation with the Cd-B and the intakes of food groups as an influencing factor for Cd exposure among the general population in South Korea. During seven Korea National Health and Nutrition Examination Surveys from 2005 to 2017, a total of 9578 individuals (4317 men and 5261 women) participated in a 24 h recall test for a dietary survey and a blood-metal survey using physical examinations performed in the same survey year. The blood Cd concentration was observed to decrease significantly (*p* < 0.05) from 1.51 µg/L in 2005 to 0.76 µg/L in 2017. In terms of the food groups, grains and cereals, potatoes and starch, and fruits were significantly correlated with the corresponding Cd concentrations and also showed decreased intakes. For Koreans, the observed decrease in blood Cd concentrations was probably caused by a decrease in the intake of food groups of plant origin.

## 1. Introduction

Cadmium (Cd) is a well-known toxic metal naturally present in the environment. Environmental exposure to toxic metals has particular importance due to the long biological half-lives of such metals. Moreover, humans are continuously exposed to such metals over their lifetimes, albeit at low concentrations [1,2,3].

Dietary intake, like background environmental exposure, can vary over time. The time-trend management of exposure to hazardous materials, in accordance with the Environmental Health Act in Korea, aims for the well-being of the general population. As components of epidemiological surveys undertaken by the South Korean government, nationwide surveys were conducted. The Korea National Health and Nutrition Examination Survey (KNHANES) is the largest epidemiological survey performed by the South Korean government on the impact of nutrition and includes a dietary survey and physical examination survey (Korea Disease Control and Prevention Agency, Ministry of Health and Welfare, South Korea, 2021). These surveys provide information on dietary intake factors and health status among the general South Korean population [4,5].

Our previous survey showed that the primary source of Cd exposure for the general population of South Korea (Korea) is oral intake, predominantly that of cooked rice, which belongs to the food group of grains and cereals and represents the main energy source of the Korean population. Similar trends were found in Japanese studies [6,7], and the intake amount for each food was found to be important for dietary Cd intake [5]. However, dietary changes have occurred over the last ~30 years among the Korean population. Daily dietary Cd intake (Cd-D) was reported to show a minor decreasing trend over the past 10 years based on the intake amount for each food group [8]. However, that previous study did not clearly identify a noteworthy reduction in corresponding blood Cd concentrations (Cd-Bs) related to the Cd-D in individuals. Therefore, determining the correlation between biological markers of exposure to Cd, such as Cd-B and Cd-D, for each food group is necessary to study the background exposure to Cd among the general population in Korea.

The present study identified the time-trend variations of Cd-B, as a biological marker of exposure to Cd, from 2005 to 2017. Using a correlation analysis, the variation in the intake (g) of Cd-B was evaluated for each influential food group.

## 2. Materials and Methods

### 2.1. Data Sources for the Study Participants

All the data sources for the study participants were downloaded freely from the online Korea National Health and Nutrition Examination Survey (KNHANES), hosted by the Korea Disease Control and Prevention Agency (KDCA) for research purposes. None of the data included private information that could identify the participants.

The study participants were subjected to KNHANES I (1998), KNHANES II (2001), KNHANES III (2005), KNHANES IV (2007–2009), KNHANES V (2010–2012), KNHANES VI (2013–2015), and KNHANES VII (2016–2018). Among the surveys, participants were selected from KNHANES III (2005), KNHANES IV (2008), KNHANES V (2011, 2012), KNHANES VI (2013), and KNHANES VII (2016, 2017). The participants participated in both a 24-h recall test, for the dietary survey, and a urinary blood-metal survey using a physical examination [9]. The number of KNHANES participants in the seven surveys from 2005 to 2017 totaled 46,818 in the dietary survey and 48,040 in the physical examination survey. The participants were reclassified based on an age range of 20–49 for the data analysis. In these reclassified results, a total of 9578 participants were selected for data analysis for both the 24-h recall test and the blood-metal survey. The participants in each survey year (Table 1) totaled 1423 in 2005 (709 men and 714 women), 1343 in 2008 (611 men and 732 women), 1368 in 2011 (623 men and 745 women), 1347 in 2012 (617 men and 730 women), 1363 in 2013 (634 men and 729 women), 1514 in 2016 (596 men and 918 women), and 1220 in 2017 (527 men, and 693 women). In the KNHANES surveys, the ages of the participants were as follows: 20.0% (N, 1919) were 20–29 years of age, 26.2% (N, 2510) were 30–39 years of age, 26.9% (N, 2579) were 40–49 years of age, and 26.8% (N, 2570) were 50–59 years of age (Figure 1). The general characteristics of the 9578 participants are shown in Figure 2. The average height was 170.0–172.8 cm for men and 158.0–159.3 cm for women, with an average of 164.0–165.1 cm for the general Korean population. The average weight was 70.2–74.3 kg for men and 57.6–59.1 kg for women, with an overall average of 63.6–65.5 kg. The average BMI was 24.1–24.8 for men, 22.9–23.4 for women, and 23.5–23.9 overall. The average total food intake based on the 24-h recall test was 1569.1–1990.8 g for men and 1234.5–1518.3 g for women, with 1406.4–1704.3 g as the overall average. The average daily energy intake was 2298.3–2561.3 Kcal for men and 1660.0–1839.8 Kcal for women, with 1950.6–2136.4 Kcal as the overall average.

### 2.2. Biological Exposure Markers and Dietary Intake in Each Food Group for Cd Intake

The Cd-B data included the biological markers of exposure to Cd for whole blood samples from the participants (Figure 3). These data for Cd-B were paired with the individual dietary intake amounts for each food group as shown in Table 1. The dietary intake amounts (g) for individual participants were calculated and divided based on food groups of plant and animal origin (Table 1). The food groups of plant origin included grains and cereals, potatoes and starch, sugar and sweets, pulses, nuts and seeds, vegetables, mushrooms, fruits, seaweed, beverages, seasonings, oils and fats (plant origin), and others (plant origin). The food groups of animal origin included meats and poultry, eggs, fish and shellfish, milk and dairy products, oils and fats (animal origin), and others (animal origin). For the dietary intake survey, the participants responded to a written questionnaire on their intake of foods based on a 24 h recall test.

### 2.3. Cd-B Analysis

Cd-B analysis were used graphite furnace atomic absorption spectrometry in each participant analysis institution. The results of Cd-B were strictly managed by the adoption of external and internal quality control programs. External quality control was evaluated by the participating authoritative agency, e.g., the G-EQUAS. Internal quality control was managed by the control guideline from KNHANES [10,11].

### 2.4. Statistical Analysis 

The IBM SPSS Statistics 26 software (Release 26, 64-bit edition, IBM Corporation, Armonk, NY, USA) was used for data classification, reorganization, and statistical analysis. The general characteristics of the participants and food groups according to a 24 h recall test are expressed in terms of the AM (arithmetic mean). A log-normal distribution was assumed, with Cd-B as the biological marker of exposure to Cd and Cd-D. The Cd levels are expressed in terms of the GM (geometric mean). GM were conducted after logarithmic transformation to LN of the Cd-D and Cd-B. ANOVA and Scheffe tests with post-hoc analysis were used to compare the means between survey-year groups. Pearson’s correlation coefficients for Cd-B and each food group, based on the survey site, were employed to determine the correlation.

## 3. Results

### 3.1. Cd-B from the 2005 to 2017 Surveys

The total Cd-B among the 9578 participants was 0.94 µg/L for the GM in the seven surveys from 2005 to 2017. The men (N: 4317) presented 0.87 µg/L (GM), while the women (5261) presented 1.00 µg/L (Figure 3). The total Cd-B (men and women) was 1.51 µg/L in the 2005 survey, 0.89 µg/L in the 2008 survey, 0.95 µg/L in the 2011 survey, 0.91 µg/L in the 2012 survey, 0.79 µg/L in the 2013 survey, 0.90 µg/L in the 2016 survey, and 0.76 µg/L in the 2017 survey. The total Cd-B (men and women) showed a decreasing tendency in the 2005 survey compared to the 2017 survey (*p* < 0.05 based on a one-way ANOVA and Scheffe test using post-hoc analysis). Cd-B showed the highest GM concentration in the 2005 survey and the lowest in the 2017 survey. Compared to the 2005 survey, the Cd-B in 2017 decreased by 50.3% from the total Cd-B, with a 40.9% decrease among men and a 58.9% decrease among women. The Scheffe test using post-hoc analysis showed a difference in the GM between survey years across all the groups (men, women, and totals). Consequently, the Cd-B of the annual survey results showed a gradual decrease in Cd-B in both men and women from 2005 to 2017 among the general population in Korea.

### 3.2. Variation of the Intake Amount in Each Food Group from 2005 to 2017

The daily food intake corresponding to each individual Cd-B value is shown in Table 1. The GM values for individual food intake corresponding to Cd-B were divided into 18 groups of plant and animal origin. As previously described, food intake was determined using a 24-h recall test. The daily food intake by food group among all 9578 subjects was 262.4 g for grains and cereals, 29.7 g for potatoes and starch, 5.3 g for sugar and sweets, 24.3 g for pulses, 1.3 g for nuts and seeds, 264.2 g for vegetables, 5.8 g for mushrooms, 127.7 g for fruits, 5.3 g for seaweed, and 24.8 g for seasonings. For food groups of animal origin, the intake of meat and poultry totaled 81.5 g, eggs totaled 27.4 g, fish and shellfish totaled 34.1 g, and milk and dairy products totaled 122.7 g. The following food groups of plant origin were the highest in terms of daily intake: grains and cereals, vegetables, and fruits. For the foods of animal origin, meats and poultry and milk and dairy products had the highest intakes.

The food groups were analyzed by a one-way ANOVA and Scheffe test for post-hoc analysis in order to identify decreasing changes in intake over a long period of time among the general Korean population. All the food groups showed significant differences between the annual surveys according to a one-way ANOVA (*p* < 0.05). The intakes of foods from plant origin (such as grains and cereals, potatoes and starch, sugar and sweets, pulses, vegetables, mushrooms, fruits, seasonings, and oils and fats (plant origin)) were significantly different between the survey years, showing a decreasing time-trend variation (*p* < 0.05 based on a Scheffe test). For the intake of animal-based foods, eggs, milk and dairy products, and oils and fats (animal origin) showed significant differences between the annual surveys. These food groups of animal origin showed decreasing trend in intake over time.

### 3.3. Correlation between Cd-B and Dairy Dietary Intake in Each Food Group 

The correlation between Cd-B and the daily dietary intake for each food group was investigated as the main influencing factor for Cd exposure from the food groups. Table 2 presents the Pearson correlation coefficient and significance for each food group for Cd-B. The correlation analysis used averages based on 31 survey areas for sampling from 17 administrative districts in Korea.

The three plant-based food groups showed significant coefficients of correlation with Cd-B, 0.5 or more, namely 0.505 for grains and cereals (*p* < 0.01), 0.519 for potatoes and starch (*p* < 0.01), and 0.563 for vegetables (*p* < 0.01), with a significant correlation between the daily food intake and Cd-B (significant decreases in both). For pulses and fruits, the Pearson correlation coefficient was lower than 0.5 (0.366, 0.405, *p* < 0.05). However, the one-way ANOVA (Table 1) showed a gradual decreasing trend by year (*p* < 0.05). There were food groups of animal origin (eggs and fish and shellfish) that were distinctly correlated with Cd-B (*p* < 0.05). The correlation coefficient was 0.367 and 0.375, indicating that food groups were correlated but with values lower than 0.5 for the Pearson correlation coefficient.

## 4. Discussion

Food and air intake account for almost all the Cd exposure among the general population [12,13]. According to previous studies, in Korea and many other countries, almost all the Cd exposure among the general population occurs through food intake [14]. The Cd-D values were evaluated via the 24-h food duplicate method using atomic absorption spectrophotometry after the wet digestion of mineral acids. Exposure to Cd from food intake mainly occurs through cooked rice, which is the most important source of Cd and the primary source of food consumed among the Korean population [13,14]. Compared to past years, however, present-day Koreans are exhibiting changes in their dietary patterns. Thus, there is a need to compare food intake patterns up to the present day. From these Cd exposure, Cd-B has been recognized as valid for recent exposure [15,16]. 

A recent Cd-B study (published in 2017) showed a significant decrease compared to the results of previous studies [17]. In 1985, Lee and Kim reported 2.82 µg/L among urban residents and 2.43 µg/L among rural residents as the GM of the Cd-B. In a 2001 survey by Kim et al., a 1.43 µg/L Cd-B was reported among men and women in the Seoul area. In a 2007 survey by Moon et al., the Cd-B was found to be 1.70 µg/L in coastal areas and 1.72 µg/L in inland areas [18]. These studies showed a 50.3% decrease in male and female populations. Men showed a 40.9% decrease, while women presented a 58.9% decrease. Thus, whether Cd-D has decreased is an important factor to consider. The Cd-D (Table 3) was calculated using the food intake amount as shown in Table 1, and the Cd contents in each food group, as shown in Table 3, were used to compare the Cd-D levels between food groups. The Cd-D from foods, such as seaweed (3.66 µg/day), grains and cereals (3.36 µg/day), fish and shellfish (3.00 µg/day), vegetables (2.11 µg/Kg), and fruits (0.80 µg/Kg), was higher among the Korean population. These daily Cd intakes were also comparatively higher than those from other food groups. Among them, grains and cereals, vegetables, and fruits of plant origin foods showed a decreasing trend over time. 

Notably, seaweed (of plant origin) and fish and shellfish (of animal origin) showed extremely high Cd contents. As the exposure limit for Cd intake, the PTMI (provisional tolerable monthly intake) outlined by the FAO/WHO of JECFA was 25 µg/Kg bw/month. Considering an average body weight of 64.4 Kg for the total AM (men and women), as shown in Table 2, the PTMI would be 1610 µg/month among the Korean population. Thus, in terms of daily intake, the PTDI corresponding to the PTMI is 53.7 µg/day. In the Korean population, four papers reported Cd contents ranging from 50 to 4730 µg/Kg in seaweed. The author then recalculated a GM of 689.9 µg/Kg from these 31 representative seaweed-Cd-content data [19,20,21,22]. Seaweed was thus considered the highest-Cd-content food group consumed among the Korean population. Seaweed is commonly consumed by the Korean population, but its intake widely varies from person to person as a favorite food. The Cd contents also have very wide ranges. The daily Cd intake from seaweed was previously estimated at 3.66 µg/day and did not exceed the PTDI (53.7 µg/day) for a single food group. However, if the Cd content of seaweed is 4730 µg/Kg, the estimated Cd intake would be 25.07 µg/day and thus accounted for PTDI by 46.7%, as shown in Table 3. Caution should thus be taken in the amount of seaweed consumed and the intervals between their intake [23]. These foods could be an important source of Cd exposure from dietary intake in Korea. However, in the present study, a correlation with the Cd-B was not observed or clear decreasing trends of Cd intake. Consequently, a clear explanation for Cd exposure cannot be found for this food group. More studies are necessary to determine the intake amounts, intake patterns, and contents of Cd.

Fish and shellfish comprise another food group shown to have a high Cd content. In the food group, fish had about 33.0 µg/Kg of Cd based on the GM recalculated from 43 cases of representative reported data, which is not comparatively high. The range was 1.0–1090 µg/Kg [19,24,25,26,27]. The GM of shellfish Cd content was 274.8 µg/Kg based on the recalculated GM, with 68–793 µg/Kg as the range among the 43 reported data sets [19,28,29]. The Cd-D from fish was recalculated as 3.00 µg/day (based on 87.9 µg/Kg and 34.1 g daily intakes), which did not exceed the PTDI. However, the consumption of 1090 µg/Kg of food with Cd-D of 37.17 µg/day would account PTDI by 69.2% for a single food group. Fish and shellfish, mollusks, and crustaceans were all shown to have high Cd contents. The Cd content in mollusks was 110.5 µg/Kg based on the recalculated GM for four cases of reported data [19,24], with a range of 35–677 µg/Kg. Crustaceans presented a Cd content of 88.1 µg/Kg based on the recalculated GM for eight cases of reported data [24,30], with a range of 22–280 µg/Kg. Therefore, the foods resulting in the highest Cd exposure among the general Korean population are, potentially, seaweed, fish and shellfish, mollusks, and crustaceans. However, these foods with high Cd contents were not found to be frequently consumed [31].
ijerph-19-00152-t003_Table 3Table 3Cd-exposure sources and probability of dietary Cd intake in Korean food.Food GroupNGM [Min–Max](µg Cd/Kg)Total GM of Daily Intake (g)Cd-D [Min–Max](µg Cd/Day)ReferencesSeaweed31689.9[50.0–4730.0]5.33.66[0.27–25.07][19,21,22,26]Grains and cereals3212.8[3.4–48.0]262.43.36[0.89–12.60][19,24,32,33]Fish and shellfish10687.9[1.0–1090.0]34.13.00[0.03–37.17][19,24,25,26,27,28,29]Vegetables238.0[0.1–30.0]264.22.11[0.03–7.93][19,24,34]Fruits106.3[2.1–14.0]127.70.80[0.27–1.79][19,35]Seasonings625.2[15.0–49.0]24.80.62[0.37–1.22][19,24]Pulses922.0[9.0–33.0]24.30.53[0.22–0.80][19,24,32]Meats and poultry36.0[0.7–24.0]81.50.49[0.06–1.96][24]Potatoes and starch415.2[9.3–19.0]29.70.45[0.28–0.56][19,24,32]Mushrooms1416.6[1.9–205.0]5.80.10[0.01–1.19][36,37]Milk and dairy products40.5[0.1–1.8]122.70.06[0.01–0.22][19,38]Nuts and seeds126.9[26.9]1.30.03[0.03][19]Eggs10.4[0.4]27.40.01[0.01][19]Sugar and sweets171.6[0.1–4.7]5.30.01[0.00–0.02][19,39]Others (plant origin)10.7[0.7]5.40.00[0.00][38]Oils and fats (plant origin)110.3[0.1–1.0]4.80.00[0.00–0.00][19,38]Total
920.3[120.8–6277.5]1026.715.25[2.48–90.57]



## 5. Conclusions

The present study identified the time trends of the Cd-B in seven nationwide surveys from 2005 to 2017 and the correlation between the Cd-B and daily intakes (g) for food groups. Exposure to Cd from food intake were identified through foods from plant origin (grains and cereal, potatoes and starches, and fruits), which is the most influential source of Cd-D among the Korean population. The Cd-B decreased significantly (*p* < 0.05) from 1.51 µg/L in 2005 to 0.76 µg/L in 2017. In terms of food groups, the daily intakes (g) of grains and cereals, potatoes and starch, and fruits were significantly decreased with the Cd-B. 

The foods resulting in the highest Cd exposure among the general Korean population are, potentially, seaweed (of plant origin), fish and shellfish, mollusks, and crustaceans (of animal origin). However, these foods were not found to be frequently consumed. They were also thought to be a potential exposure factor due to exceeding the PTDI (53.7 µg/day). Caution should thus be taken concerning the amount consumed and the intervals between intakes.

## Figures and Tables

**Figure 1 ijerph-19-00152-f001:**
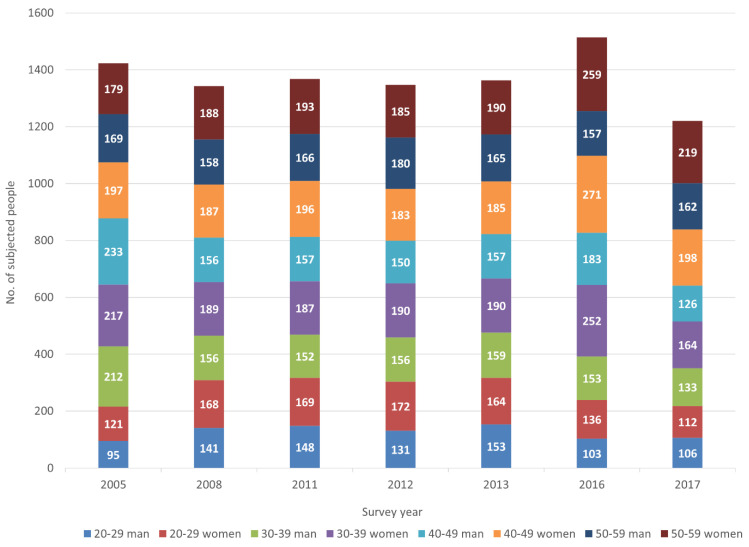
Numbers of subjects with blood Cd concentrations and dietary intake data from the 2005 to 2017 surveys.

**Figure 2 ijerph-19-00152-f002:**
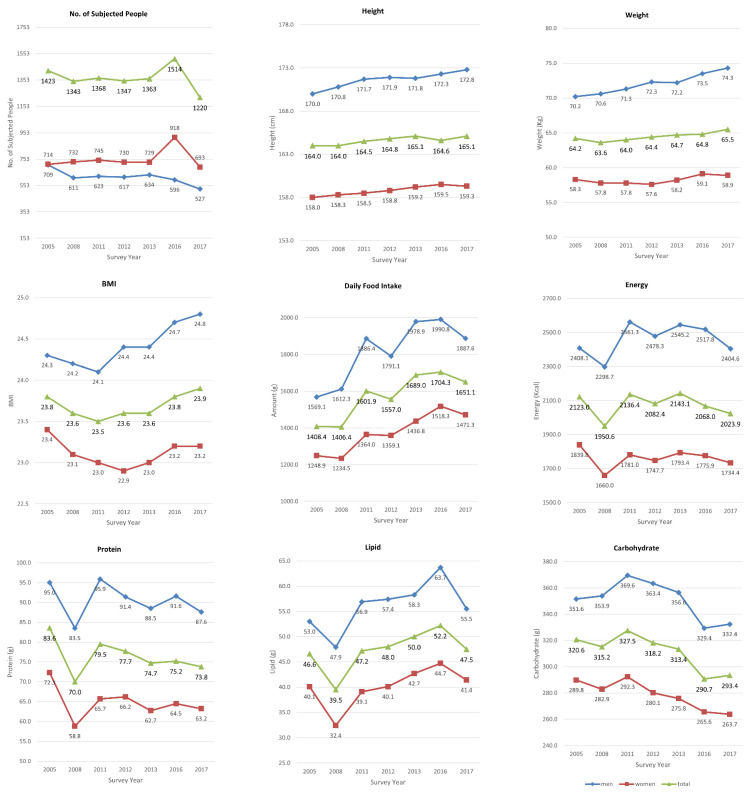
General characteristics of the subjects with AM from the 2005 to 2017 surveys.

**Figure 3 ijerph-19-00152-f003:**
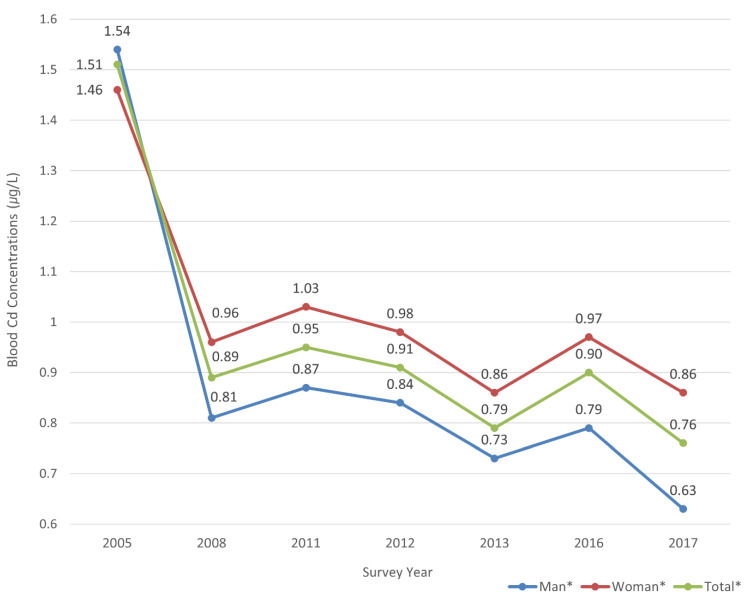
Variation of blood Cd concentrations (µg/L) from the 2005 to 2017 surveys. * *p* < 0.05 using a one-way ANOVA; a Scheffe test was used to compare the 2005 survey with the other survey. All groups (total, men, and women) were significant (*p* < 0.05), with the 2005 survey using a Scheffe test employed as the post-hoc analysis.

**Table 1 ijerph-19-00152-t001:** Variation GM (g) in each food group based on a 24-h recall test from the 2005 to 2017 surveys.

Food Group	2005 (1423) ^†^	2008 (1343)	2011 (1368)	2012 (1347)	2013 (1363)	2016 (1514)	2017 (1220)	Total (9578)
*Plant origin*								
Grains and cereals *	**295.9**	267.7	**275.9**	**265.1**	**257.2**	**239.8**	**244.7**	262.4
Potatoes and starch *	**34.8**	**45.2**	**31.8**	**27.4**	**27.1**	**25.3**	**25.8**	29.7
Sugar and sweets *	6.3	5.2	5.0	5.1	5.6	5.1	5.0	5.3
Pulses *	**37.0**	30.0	27.7	25.3	**22.2**	**16.9**	**16.0**	24.3
Nuts and seeds *	2.4	**0.9**	1.2	1.2	**1.2**	1.2	1.2	1.3
Vegetables *	**327.0**	**270.4**	**267.7**	**259.8**	**249.6**	**239.8**	**237.5**	264.2
Mushrooms *	**12.8**	8.9	**6.0**	6.4	**5.1**	**4.4**	**4.1**	5.8
Fruits *	**131.6**	**175.9**	125.2	134.3	126.5	134.3	82.3	127.7
Seaweed *	**5.6**	**4.8**	**3.1**	**3.3**	**6.0**	**6.9**	**8.8**	5.3
Seasonings *	**30.0**	**22.2**	**25.0**	**23.1**	**25.0**	**25.0**	**23.1**	24.8
Oils and fats (plant origin) *	**5.4**	4.6	5.1	4.8	5.1	**4.5**	**4.3**	4.8
Others (plant origin) *	1.3	**1.5**	**36.6**	**16.8**	5.0	7.4	7.2	5.4
*Animal origin*								
Meats and poultry *	**83.1**	**66.7**	82.3	82.3	83.1	83.1	87.4	81.5
Eggs *	**33.8**	26.8	27.9	**28.5**	**25.5**	**27.7**	**23.3**	27.4
Fish and shellfish *	**41.3**	**29.7**	29.7	25.5	32.5	**38.9**	**46.1**	34.1
Milk and dairy products *	**145.5**	159.2	130.3	131.6	127.7	**93.7**	106.7	122.7
Oils and fats (animal origin) *	**6.2**	1.5	**1.6**	**1.6**	**1.3**	**0.6**	**0.7**	1.9
Others (animal origin) *	1.7	11.4	0.9	0.4	12.6	4.5	6.0	1.0

* *p* < 0.05 using a one-way ANOVA. Bold font indicates significant results (*p* < 0.05), with the 2005 or 2008 survey using a Scheffe test employed as the post-hoc analysis. ^†^ Numbers in parenthesis are numbers of participants.

**Table 2 ijerph-19-00152-t002:** Pearson’s coefficients of correlation between blood Cd concentrations and food groups based on the survey site (N: 31).

	Correlation Coefficient	*P*
*Plant origin*		
Grains and cereals	0.505 **	0.004
Potatoes and starch	0.519 **	0.003
Sugar and sweets	0.113	0.545
Pulses	0.366 *	0.043
Nuts and seeds	0.049	0.794
Vegetables	0.563 **	0.001
Mushrooms	0.114	0.543
Fruits	0.405 *	0.024
Seaweed	0.333	0.067
Seasonings	0.303	0.098
Oils and fats (plant origin)	0.084	0.651
Others (plant origin)	−0.014	0.942
*Animal origin*		
Meats and poultry	0.294	0.108
Eggs	0.367 *	0.042
Fish and shellfish	0.376 *	0.037
Milk and dairy products	0.316	0.084
Oils and fats (animal origin)	0.069	0.717

* *p* < 0.05, ** *p* < 0.01.

## Data Availability

KDCA Korea National Health and Nutrition Examination Survey (KNHANES). Available online: https://knhanes.kdca.go.kr/knhanes/eng/index (accessed on 5 November 2021).

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
