# Peer review of "Blood Concentrations and Dietary Intake of Cd among the General Population in South Korea"

_ijerph, 2021, doi:10.3390/ijerph19010152_

Round 1
Reviewer 1 Report
Ok. The paper was significantly improved. table 3 is still a mess but probably the fault is the pdf? please check. then it is ok, for me
Author Response
Reviewer 1
Comment 1
Table 3 is still a mess but probably the fault is the pdf? please check.
Answer
Table 3 was revised to Figure 3.
Comment
English language and style
Answer
All sentences were checked and modified and marked in red.

Reviewer 2 Report
Blood concentrations and dietary intake of Cd among the general population in South Korea
My decision: Rejected
This paper investigated and studied the relationship between the content of cadmium in blood and the types and intake of food in Korean people, determined the time trend of the change of the content of cadmium in blood, and found out the factors affecting the exposure of Cd in the Korean general population. The author has done a lot of related data survey work. However, there are still many problems to be solved. In the current version, I do not recommend publishing.
- Table 3 table column width setting inappropriate, it is suggested that the column width adjustment form, the data content and reasonable arrangement, avoid to cause readers confused.
- The meaning of ' * ' is explained in the footnote to table 3, but the meaning of ' * * ' in table 5 is not clear. Please provide additional explanation.
- The number of decimal points in Table 6 remains inconsistent, and the table layout is unreasonable. It is suggested to modify and adjust. Please examine the full text carefully to avoid similar situations.
- In this paper, the author makes a classified statistics on the gender, age and living area of the survey population. The survey results show that different dietary intake schemes have different effects on the change of blood cadmium content, but there is a lack of detailed exploration of its causes. In addition to the types and contents of food intake, are the reasons for the differences related to gender and age of the surveyed population? How is it related?
- 'GM' and 'GSD' are mentioned in this paper. Please provide the specific calculation formula.
- The font size and line spacing of the text section should be adjusted to maintain consistency.
- There are many tables in the text, so it is suggested to use Figure appropriately to make the results clearer and more intuitive for readers to read.
- In the summary part, it is suggested to refine innovation points so as to highlight the significance and importance of research.
- There is no source or basis for how much cadmium is in food and how it is obtained.
- The author pointed out that there was a positive correlation between some food types and blood cadmium concentration. The correlation was explained only by intake, the lack of in-depth analysis of the causes and the consideration of other factors.
Author Response
Reviewer 2
Comment 1
Table 3 table column width setting inappropriate, it is suggested that the column width adjustment form, the data content and reasonable arrangement, avoid to cause readers confused.
Answer
Table 3 was revised to Figure 3 and indicated in red.
Comment 2
The meaning of ' * ' is explained in the footnote to table 3, but the meaning of ' * * ' in table 5 is not clear. Please provide additional explanation.
Answer
Table 5 was revised and indicated in red.
Comment 3
The number of decimal points in Table 6 remains inconsistent, and the table layout is unreasonable. It is suggested to modify and adjust. Please examine the full text carefully to avoid similar situations.
Answer
Table 6 was revised and indicated in red.
Comment 4.
In this paper, the author makes a classified statistics on the gender, age and living area of the survey population. The survey results show that different dietary intake schemes have different effects on the change of blood cadmium content, but there is a lack of detailed exploration of its causes. In addition to the types and contents of food intake, are the reasons for the differences related to gender and age of the surveyed population? How is it related?
Answer
This is the author's opinion on this comment. The overall core content of this paper is that the amount of cadmium consumed by all Koreans has decreased. This study was conducted based on the results of my previous studies showing differences by gender, age, and region in papers related to cadmium intake by the general population. This difference has been confirmed not only in Korea but also in Japan. However, cadmium exposure in the general population is a major source of exposure to food, and cadmium present in the soil is presumed to be a major factor for Asians who mainly eat grains and vegetables, not meat.
In the case of the general population as well as the surveyed population, there are naturally differences depending on gender, age, and region. In short, men and women differ in daily food intake. This is when the region and age are equal. When regions are different, that is, when living inland and coastal, there is a difference in cadmium intake depending on the region. In other words, it is reported that the intake of seafood naturally increases when living on the coast, and as a result, the blood concentration of cadmium increases. The same is true of age. In Korea, meal patterns are also quite stable in the age group in their 40s. However, in the case of those in their 20s and 30s, the amount of food consumed per day is large and the amount of activity is high, so eating out is frequent, which can lead to variations in cadmium intake.
Comment 5
'GM' and 'GSD' are mentioned in this paper. Please provide the specific calculation formula.
Answer
‘2.4. Statistical analysis’ was revised and indicated in red.
Comment 6
The font size and line spacing of the text section should be adjusted to maintain consistency.
Answer
The entire font size and line spacing of the paper were checked and adjusted.
Comment 7
There are many tables in the text, so it is suggested to use Figure appropriately to make the results clearer and more intuitive for readers to read.
Answer
Table 1, 2, and 3 were revised to Figure 1, 2, and 3.
Comment 8
In the summary part, it is suggested to refine innovation points so as to highlight the significance and importance of research.
Answer
‘Abstract’ and ‘Conclusion’ were revised and indicated in red.
Comment 9
There is no source or basis for how much cadmium is in food and how it is obtained.
Answer
Please refer to Table 6. Table 6 shows the geometric mean and minimum maximum of each food group data by collecting reports on cadmium for each food group consumed by Koreans, and based on this data, the daily intake was calculated.
Comment 10
The author pointed out that there was a positive correlation between some food types and blood cadmium concentration. The correlation was explained only by intake, the lack of in-depth analysis of the causes and the consideration of other factors.
Answer
As previous many studies, the characteristic of cadmium exposure in the general population is from food intake. Respiratory or Skin absorption was too low.
Comment
English language and style
Answer
All sentences were checked and modified and marked in red.

Round 2
Reviewer 2 Report
Blood concentrations and dietary intake of Cd among the general population in South Korea
My decision: Accept
The author revised and improved the article carefully according to the requirements of the reviewers, and did a lot of work. The readability of the article has been significantly improved. I am satisfied with the revised version and the author’s reply. I think it can be accepted and published.
This manuscript is a resubmission of an earlier submission. The following is a list of the peer review reports and author responses from that submission.
Round 1
Reviewer 1 Report
I think this paper reports a good work on relationship between the Cd-intake and human-being's health. The data are sound, and the paper is well organized. It can be considered to be accepted for publication in this journal after rewrite the conclusion. I suggest the author rewrite the conclusion by summarizing the changes of Cd-intake for different people groups for different food intake, and stressing the possible relationship between the Cd-intake and human-being's health.
Reviewer 2 Report
The aim of the study was to identify the trends of blood Cd concentrations and its determinants relevant to dietary among the general population in South Korea. There are several main issues that should be concerned. First, current statistical analysis based on scheffe test by the post-hoc analysis cannot efficiently evaluate the trends of blood Cd concentrations, as well as the trends of different dietary intakes. P values for trend with linear function are needed to fit the overall trends across seven survey years. Second, although dietary intake was evaluated by a 24 h recall test, the corresponding arithmetic mean of each food was essentially used to measure cumulative intake from 2005 to 2017. However, blood Cd is a biomarker for recent exposure, could insufficiently be used as an effect biomarker in response to cumulative exposure. Urine Cd could be considered instead. Third, lab information including measures of Cd concentrations, the limit of detection, and so on, should be briefly added to the section of Materials and Methods. Additionally, it might be better to move the Table 6 to the section of Results.
Reviewer 3 Report
The work deals with cadmium contamination in a Korean population. As stated by the same author, line 192, a similar work has already been published in 2017. And the trend seems identical: cadmium levels are falling and, as expected, there is a correlation with the diet.
Regarding the diet, attention is paid to different foods but nothing is known about their origin and levels of contamination. Is the population coming from the same area? Is there a similar correlation in rural and urban populations? between those who eat self-produced foods and those who rely on large shops for food?
Apart from these considerations, data are exposed in a heavy and difficult to read manner. The results are lost in a sea of ​​apparently insignificant digits.